# Novel Virus Air Sampler Based on Electrostatic Precipitation and Air Sampling of SARS-CoV-2

**DOI:** 10.3390/microorganisms11040944

**Published:** 2023-04-04

**Authors:** Kyohei Fukuda, Hiroaki Baba, Mie Yoshida, Kouichi Kitabayashi, Shinjirou Katsushima, Hiroki Sonehara, Kazue Mizuno, Hajime Kanamori, Koichi Tokuda, Atsuhiro Nakagawa, Akira Mizuno

**Affiliations:** 1AMANO Co., Ltd., 275 Mamedocho, Kohoku-ku, Yokohama 222-8558, Japan; 2Department of Infectious Diseases, Internal Medicine, Tohoku University Graduate School of Medicine, 2-1 Seiryo-machi, Aoba-ku, Sendai 980-8574, Japan; 3Genome Clinic Co., Ltd., 2-5-1 Chuo, Chuo-ku, Chiba 260-0013, Japan; 4MRC LLC, 5-29-12, Hongo, Bunkyo-ku, Tokyo 113-0033, Japan; 5Preferred Networks, Inc., 1-6-1 Otemachi Building, Otemachi, Chiyoda-ku, Tokyo 100-0004, Japan; 6Department of Neurosurgery, Tohoku University Graduate School of Medicine, 2-1 Seiryo-machi, Aoba-ku, Sendai 980-8574, Japan

**Keywords:** electrostatic precipitator, air sampler, SARS-CoV-2

## Abstract

The assessment of airborne viruses in air is a critical step in the design of appropriate prevention and control measures. Hence, herein, we developed a novel wet-type electrostatic air sampler using a viral dissolution buffer containing a radical scavenging agent, and verified the concentration of severe acute respiratory syndrome coronavirus 2 (SARS-CoV-2) RNA in the air of hospital rooms inhabiting coronavirus disease 2019 (COVID-19) patients and public areas. RNA damage caused by corona discharge was negligible when Buffer AVL was used as the collecting electrode. The viral RNA concentration in the air of the room varied by patient: 3.9 × 10^3^ copy/m^3^ on the 10th day after onset in a mild case and 1.3 × 10^3^ copy/m^3^ on the 18th day in a severe case. Viral RNA levels were 7.8 × 10^2^ and 1.9 × 10^2^ copy/m^3^ in the air of the office and food court, respectively, where people removed their masks when eating and talking, but it remained undetected in the station corridor where all the people were wearing masks. The assessment of airborne SARS-CoV-2 RNA using the proposed sampler can serve as a basis for the safe discontinuation of COVID-19 isolation precautions to identify exposure hotspots and alert individuals at increased infection risks.

## 1. Introduction

The global spread of severe acute respiratory syndrome coronavirus 2 (SARS-CoV-2), which causes coronavirus disease 2019 (COVID-19), has resulted in a global public health crisis [1]. SARS-CoV-2 can be transmitted through the air via aerosol particles emitted by coughing, sneezing, or breathing [2,3,4]. The assessment of airborne viruses in the air is a critical step toward understanding the transmission behavior and designing the appropriate prevention and control measures; however, there is no standard protocol for air sampling [5].

An appropriate, easy-to-use sampler is necessary to obtain more accurate data of airborne transmission. Currently, impingers and filter-type air samplers are generally used to measure virus concentrations in the air [6,7,8]; however, there are drawbacks. In particular, the collection efficiency of impingers is significantly low, and post-treatments, such as the elution of collected viruses, are necessary for filter-type samplers that use gelatin filters. In addition, high-pressure air pumps with loud operation noises are required for impingers and filter-type air samplers, because the pressure drops of these samplers are larger than those of electrostatic samplers.

Electrostatic precipitation has been extensively used for industrial air cleaning [9]. The sampling of bio-particles using electrostatic precipitation has advantages such as low pressure drops, high collection efficiencies for fine particles [10], and collection into liquid for smooth processing, to conduct polymerase chain reaction (PCR) detection [9,11,12,13]. However, corona discharge that occurs in electrostatic precipitation can damage the viral nucleic acid due to oxidative radicals [8,14,15,16,17].

In this study, we developed a novel wet-type electrostatic sampler using a viral dissolution buffer containing a radical scavenging agent as a collecting electrode instead of a dry metal electrode, and used it to verify the concentration of SARS-CoV-2 RNA in the air of a hospital room inhabiting COVID-19 patients, in addition to public areas. The objective was to establish the efficacy of the developed sampler in enabling highly efficient, simple, and low-noise sampling, in addition to its potential contribution to an improved understanding of SARS-CoV-2 viral shedding, and to enhance surveillance efforts in the current global COVID-19 pandemic.

## 2. Material and Methods

### 2.1. Structure and Performance of the Developed Air Sampler

The wet-type electrostatic precipitator air sampler consisted of an electrostatic sampling unit, a blower-type fan, and a battery-operated high voltage power supply (Figure 1A). The size of it was 300 mm × 300 mm × 150 mm. The electrostatic sampling unit comprised a commercially available vial bottle with an inner diameter of 40 mm, and insulating chloroprene rubber cover with a stainless tube at the center (diameter of 20 mm) and 10 holes arranged around it (Figure 1B). The vial bottle contained 2.9 mL of liquid electrode comprising 1.12 mL of viral lysis buffer (Buffer AVL, QIAGEN, Tokyo, Japan) that contained guanidinium thiocyanate at a concentration of 50–70%, and 1.78 mL of distilled water (DW). The conductivity of the Buffer AVL ranged from 41.49 to 45.52 mS/cm (1/10 dilution), and the pH ranged from 7.30 to 7.50. Ten corona discharge electrodes made of a bundle of fine fibers of stainless steel (100 fibers with fiber diameters of 12 µm) [13] were attached at the tip of the stainless tube. 

A blower-type fan with an operating voltage of 5 V and a current of 0.4 A was placed at the outlet of the vial bottle, with a minimum diameter of 10 mm. The pressure drop of the system was 11 Pa, and the operating noise of the air sampler was 31 dB. Air that flowed into the vial bottle through the stainless tube of the cover, was exposed to the corona discharge, and then exhausted through the holes of the bottle cover.

### 2.2. Measurement of the Particle Collection Efficiency of the Developed Air Sampler When Compared with an Impinger and a Filter-Type Air Sampler

To determine the optimal air sampler flow rate, the particle collection efficiency was measured three times for each flow rate of 10 L/min, 20 L/min, and 40 L/min, at room temperature of 20 °C and a relative humidity level of 50%. The corona discharge was unstable due to the roughness of the liquid surface, as induced by the air flow when the flow rate was over 40 L/min. A high voltage of −6 kV was applied to the corona discharge electrodes. The discharge current was measured at the beginning and end of the sampling period. A particle counter (Model 3888, Kanomax Japan Inc., Osaka, Japan) was connected to the developed air sampler to count the number of suspended particles in the air (Figure 1B). Given that the size of the SARS-CoV-2 particles released from the human body and present in the air is ≤1.0 μm [18], the target size of the particles was set at 0.3–0.5 μm. The collection efficiency was determined by comparing the counts with and without corona discharges. The following equation was used to calculate the collection efficiency.
$$H\;\left(\%\right)=\left(1-\frac{count\;of\;the\;particle\;at\;the\;outlet\;with\;corona\;discharge}{count\;of\;the\;particle\;at\;the\;outlet\;without\;corona\;discgarge}\right)\times 100$$

As a comparison, the particle collection efficiencies of an impinger and a gelatin filter-type sampler were measured three times each. The SPC Midget Impinger Type G-1 Single (Shibata Scientific Technology Ltd., Saitama, Japan) was used with a solution volume of 20 mL and a flow rate of 5 L/min. Three impingers were connected in series to measure the collection efficiency, because the efficiency of the single impinger was very low. The gelatin filter-type sampler was the MD8 Airport (Sartorius AG, Göttingen, Germany) with a flow rate of 40 L/min. Both samplers were used according to the standards of the Japan Electrical Manufacturers’ Association [19]. The following equation was used to calculate the collection efficiency.
$$\eta \;\left(\%\right)=\left(1-\frac{count\;of\;the\;particle\;at\;the\;outlet}{count\;of\;the\;particle\;at\;the\;inlet}\right)\times 100$$

### 2.3. Evaluation of the Protective Effect of the Viral Lysis Buffer on RNA from Damage Caused by Exposure to Corona Discharge

We evaluated the protective effect of the viral lysis buffer on RNA from the damage caused by exposure to corona discharge by comparing the real-time quantitative PCR (RT-qPCR) threshold (Ct) values of bacteriophage RNA in the Buffer AVL before and after exposure to the corona discharge as follows: (1) the mixture of bacteriophage MS2 (NBRC 102619) and *Escherichia coli* (NBRC 13965) was incubated overnight at 35 °C in a liquid LB medium. (2) Thereafter, 4 mL of the SM buffer and 40 µL of chloroform were added to the mixture, which was set aside at 25 °C for 30 min. (3) The bacteriophage MS2 suspension with a concentration of 5 × 10^7^ PFU/µL, as obtained by filtering the mixture with a pore size of 0.22 µm, was added to 2.9 mL of the viral lysis buffer comprising 1.12 mL of Buffer AVL and 1.78 mL of distilled water, to prevent salt deposition caused by evaporation during exposure. (4) Bacteriophage MS2 RNA was extracted from the mixture before and after exposure to corona discharge in the proposed sampler for 90 min, using the QIAamp Viral RNA mini kit (QIAGEN, Tokyo, Japan), and RT-qPCR was performed (Appendix A). The Ct values of bacteriophage MS2 RNA before and after exposure to the corona discharge were measured five times each and compared using the Student’s *t*-test.

### 2.4. Measurement of the Concentration of SARS-CoV-2 RNA in the Air of Hospital Rooms Inhabiting COVID-19 Patients and Public Spaces

The proposed air sampler was used to measure the concentration of SARS-CoV-2 RNA in the air of hospital rooms inhabiting COVID-19 patients. The target patients were a total of 10 patients hospitalized for COVID-19 and diagnosed between February and August, 2022, by positive SARS-CoV-2 PCR tests based on the nasopharyngeal swab fluid. All participants provided verbal informed consent after being informed of the study and the methodology employed for sampling. All 10 patients underwent one-step RT-qPCR SARS-CoV-2 testing of nasopharyngeal swabs with GeneXpert^®^ Xpress SARS-CoV-2 (Cepheid, Sunnyvale, CA, USA) upon admission, and the Ct values were calculated [20]. The sequencing analysis of the S1 region was performed using the Sanger method [21], and lineages were determined based on the sequence data using the Nextclade web application [22]. Patient information, such as the patient age, gender, vaccination history, and outcome, were collected from the electronic medical record. Patients were classified into three severity levels based on their worst respiratory status during the course, as follows: (1) mild: patients who did not require oxygen administration; (2) moderate: patients who required oxygen administration, but not ventilatory support; (3) severe: patients who required any ventilatory support. All patients were admitted to the same single-patient room with dimensions of 4.5 m × 3.2 m × 2.5 m (length × width × height) and an interior volume of 36 m^3^ (Figure 2) at different time periods. The sampler was placed 1.8 m from the head of the patient and 0.83 m above the floor. Sampling was performed for 90 min for each patient. Patients were informed that sampling could be stopped immediately if noise or other factors interfered with their hospital stay.

The proposed air sampler was used to measure the concentration of SARS-CoV-2 RNA in the air of public spaces, including an office room during the working hours, the food court of a shopping mall during lunch time on a holiday, and a station corridor during morning and evening commuting hours. For measurements in public places, three samplers were used simultaneously. The volumes of the office room and food court were 340 m^3^ and 2000 m^3^, respectively, and contained approximately 30 and 300 people, respectively (11 m^3^/people and 6.7 m^3^/people, respectively), who removed their masks when eating, drinking, and talking. The station corridor was 1500 m^3^ in size and contained 100 people (15 m^3^/people) all wearing masks, who were walking during sampling. The sampling was performed from 12 to 18 September 2021, when the proportion of new COVID-19 patients was five per 100,000 people in the region [23].

Viral RNA was extracted from the collected samples using QIAamp Viral RNA mini kit, and RT-qPCR was performed targeting the N2 region (Appendix A). To generate positive controls for the calibration curve, SARS-CoV-2 RNA fragments were PCR-amplified from 5.0 copies, 5.0 × 10^1^ copies, 5.0 × 10^2^ copies, and 5.0 × 10^3^ copies. Based on the calibration curve, the amount of RNA in the PCR tube was calculated, and the concentration of viral RNA per 1 m^3^ of sampled air was determined using the following equation:$$\mathrm{Cv}\;\left(copies/m^{3}\right)=\frac{N_{RNA}\times Q_{extract}}{Q\times V_{Air}}\times \frac{100}{\eta }$$

*N_RNA_*: Number of RNA fragments in the PCR tube (copies/tube)

*Q_extract_*: Volume of the extract (µL)

*Q_PCR_*: Volume of sample added to PCR tube (µL/tube)

*V_Air_*: Volume of sampled air (m^3^)

η: collection efficiency of particles between 0.3 and 0.5 µm diameter (%)

## 3. Results

### 3.1. Particle Collection Efficiency of the Developed Electrostatic Air Sampler Compared with That of an Impinger and a Filter-Type Air Sampler

At the flow rates of 10 L/min, 20 L/min, and 40 L/min, the average collection efficiencies of the proposed electrostatic air sampler were 85% (SD 2%), 63% (SD 3%), and 46% (SD 4%), respectively. The flow rate × collection efficiencies, which are proportional to the number of particles collected, were 850, 1260, and 1840, respectively, with the maximum at a flow rate of 40 L/min. Therefore, the measurement of SARS-CoV-2 RNA concentration in the air in this study was performed at a flow rate of 40 L/min. The discharge current decreased from 65 μA initially to 50 μA at the end of the sampling period. The collection efficiencies of the impinger and the gelatin filter-type sampler were 11% (SD 8%) and 97% (SD 1%), respectively.

### 3.2. Evaluation of the Protective Effect of the Viral Lysis Buffer on RNA from Damage Caused by Exposure to Corona Discharge

Figure 3 presents the RT-qPCR Ct values of bacteriophage M2 RNA in Buffer AVL before and after exposure to corona discharge. The Ct values of 25.69, 25.76, 25.6, 25.42, and 25.26 (mean [SD] of 25.5 [0.2]) after exposure were not significantly different from those of 25.43, 25.28, 25.76, 25.06 and 25.32 (mean [SD] of 25.4 [0.3]) before exposure (*p* > 0.05).

### 3.3. Measurement of the of SARS-CoV-2 RNA Concentration in the Air of Hospital Rooms Inhabiting COVID-19 Patients and Public Spaces

Patient information and the concentration of SARS-CoV-2 RNA in the air of the hospital rooms of the patients are summarized in Table 1. No patients requested for sampling to be stopped during the sampling process. SARS-CoV-2 RNA was detected in the air of hospital rooms of all patients, including patient no. 10, who had a negative nasopharyngeal SARS-CoV-2 PCR result upon admission. Virus concentrations varied from 3.0 × 10^2^ copies/m^3^ in patient no. 6 (moderate case) with Omicron 21K/Ba2, to 1.1 × 10^3^ copies/m^3^ in patient no. 2 (severe case) with Delta 21A on day 2, 1.0 × 10^3^ copies/m^3^ in patient no. 5 (moderate case), to 2.4 × 10^4^ copies/m^3^ in patient no. 8 (mild case) with both Omicron 21K/Ba1.1 on day 3, 0 copies/m^3^ in patient no. 7 (moderate case) with Omicron BA.5, to 1.7 × 10^3^ copies/m^3^ in patient no. 8 (mild case) on day 4, 0 copies/m^3^ in patient no. 8 (mild case), to 4.7 × 10^3^ copies/m^3^ in patient no. 4 (moderate case) with Omicron 21K/Ba1.1 on day 5, and 0 copies/m^3^ in patient no. 5 (moderate case), to 1.2 × 10^3^ copies/m^3^ in patient no. 6 (moderate case) on day 6 after symptom onset. Among patients with Omicron 21K/Ba1.1 on day 5, the viral concentration in the air of the hospital room of unvaccinated patient no. 9 (mild case) (1.0 × 10^2^ copies/m^3^) was lower than those of patients no. 4, 5, and 6 (moderate case) who received two and three vaccine doses, respectively (4.7 × 10^3^ copies/m^3^, 1.0 × 10^3^ copies/m^3^, and 9.2 × 10^2^ copies/m^3^, respectively). Viral RNA at a concentration of 1.3 × 10^3^ copies was detected in the hospital room of patient no. 1, who was a man in his 80s with a severe case of Omicron 21K/Ba1.1 on day 18, i.e., the day prior to his death. 

Table 2 presents the sampling results for the public places. In particular, SARS-CoV-2 RNA was detected at 7.8 × 10^2^ and 1.9 × 10^2^ copy/m^3^ in the air of the office and food court, respectively, and it was not detected in the station corridor.

## 4. Discussion

The novel wet-type electrostatic air sampler using a viral dissolution buffer as a collecting electrode could detect SARS-CoV-2 RNA in the air of a hospital room inhabiting COVID-19 patients and public areas. This study demonstrated that the wet-type electrostatic air sampler is useful for sampling SARS-CoV-2 in the air.

In this study, there was no significant increase in the RT-qPCR Ct values of bacteriophage MS2 RNA in Buffer AVL before and after exposure to the corona discharge in the proposed sampler, thus suggesting that RNA damage caused by radicals generated by the corona discharge was negligible when Buffer AVL was used as a collecting electrode in the proposed sampler. The Buffer AVL contains guanidinium thiocyanate, which acts as an antioxidant against radicals [24]. The radicals produced by corona discharge were removed via the reaction with guanidinium thiocyanate. Buffer AVL contains ribonuclease (RNase) inhibitor, which prevents RNA degradation by RNase in airborne dust and pollen that may contaminate the sampled air [25]. Therefore, RNase may also be collected by the sampler; however, the Buffer AVL allows for the collected viral RNA to be maintained.

Although the impinger sampler had a particle collection efficiency of only 11%, the wet-type electrostatic air sampler demonstrated a maximum particle collection efficiency of 85% at a flow rate of 10 L/min, which is the second highest particle collection efficiency after that of the gelatin filter sampler at 97%. In addition, the wet-type electrostatic sampler directly collects viruses in the buffer and was convenient for PCR detection. A wet-type electrostatic air sampler was previously developed and used to sample influenza viruses in the air at high efficiencies; however, the sampler maximum collection efficiency was reported to be only 47%, and its low flow rate of 6.8 L/min makes it difficult to measure viral RNA in the air of hospital rooms and public areas [26].

The observed decrease in the discharge current from the beginning to the end of the sampling period may have resulted in a reduction in the particle collection efficiency. This decrease in the current may have been caused by the reduction of liquid electrode volume and level due to spontaneous evaporation and an increase in the separation distance between the discharge electrode tip and the liquid surface. Evaporation rates increase in environments with high temperature and low humidity, leading to a decrease in collection efficiency. Controlling liquid electrode evaporation during sampling and maintaining a consistent liquid level pose potential future challenges for the proposed sampler.

The proposed sampler has a low pressure drop of 11 Pa, given that the minimum diameter of the air duct was 10 mm at the air outlet, thus allowing for air suction using a small 5 V/0.4 A blower-type fan with an operating noise of 31 dB, which is almost the same as the WHO recommended noise level of 30 dB for bedrooms [27]. Moreover, this value is lower than those of the alternative impinger-type biosampler (SKC Inc, Eighty Four, Pennsylvania, USA) and the MD8 air sampler with a gelatin filter, which exhibit noise levels of 61–66 dB [28,29] and 48 dB [30], respectively. Notably, no patients in this study complained about the noise during sampling. The proposed sampler enabled sampling of SARS-CoV-2 RNA in the hospital room air of patients undergoing treatment, without disturbing the calmness and recovery of patients.

The proposed sampler could detect SARS-CoV-2 RNA in the air of the hospital rooms of all the COVID-19 patients at concentrations ranging from 1.3 × 10^2^ to 2.4 × 10^4^ copy/m^3^, which is consistent with previously reported concentrations of 2.0 × 10^3^ to 5.2 × 10^4^ copy/m^3^ in the air of patient rooms [31,32,33]. However, the proposed sampler revealed that the concentration of RNA varied with respect to the patient and the timing of measurements. Previous reports indicated that the number of viruses in the breathed air sample and nasopharyngeal samples peaked at days 2–3 after onset [34,35]. Severe cases of COVID-19 have higher nasopharyngeal viral loads and longer periods of viral shedding than the mild/moderate cases [36]; however, viral loads and shedding dynamics are influenced by host factors, including patient age and sex, individual susceptibility and immunity from previous infections or vaccination, patient symptoms, and protective measures, including mask wearing [37]. In the hospital room air of a male patient in his 80s (patient no.1, severe case) with Omicron 21K/Ba1.1 and an unknown vaccination history, the RNA of SARS-CoV-2 was detected at 1.3 × 10^3^ copy/m^3^ until the 18th day after onset, when the patient passed away. By contrast, in the air around an unvaccinated female patient in her 30s (patient no. 10, mild case), viral RNA was detected at 3.9 × 10^3^ copy/m^3^ on the 10th day after onset, despite negative nasopharyngeal PCR testing. Given that airborne SARS-CoV-2 RNA shedding can be prolonged regardless of the severity of the patient illness, and airborne viral RNA shedding can continue even when nasopharyngeal PCR is negative, the assessment of airborne SARS-CoV-2 RNA using the proposed sampler can serve as a basis for the safe discontinuation of COVID-19 isolation precautions.

In public spaces, the proposed sampler detected RNA in the office and food court, but not in the station corridor. This is partly because all the people in the station corridor were wearing masks that inhibited the release of aerosol particles containing SARS-CoV-2 RNA into the air [38], whereas those in the office and food court removed their masks when eating, drinking, and talking [39]. Conte et al. reported that no airborne viral RNA was detected in food courts and train stations in major Italian cities during the second wave of the COVID-19 epidemic, during which physical distancing and the wearing of masks were mandatory [40]. Given that SARS-CoV-2 RNA concentrations in the air of the office and food court were calculated as 7.8 × 10^2^ and 1.9 × 10^2^ copy/m^3^, respectively, breathing at 10 L/min in these locations would result in the inhalation of over 1000 viral copies within several hours, and could result in an infection risk [41]. The assessment of airborne viral RNA in public spaces using the proposed sampler is a promising method for identifying exposure hotspots, and could be used to alert individuals at increased risks of infection to practice infection prevention measures. Furthermore, it can supplement other public health surveillance strategies, including contact tracing.

A major drawback of the proposed sampler is the use of the RT-qPCR to detect viral RNA, which fails to determine virus infectivity, because PCR can detect both live viral and non-infectious viral RNA [42]. Although viral cultures are the gold standard to determine viral infectivity and activity, they are time-consuming and labor-intensive, require a biosafety Level 3 laboratory, and cannot be implemented in clinical settings [43]. Previous studies revealed that RT-qPCR can be a surrogate for live viruses [44]. More rapid viral RNA detection methods were recently developed [45], which may allow for more real-time assessments of viral RNA in the air.

## 5. Conclusions

The novel wet-type electrostatic air sampler using a viral dissolution buffer containing radical scavenging agent, as developed in this study, can efficiently verify the concentration of SARS-CoV-2 RNA in the air of a hospital room inhabiting COVID-19 patients and public areas via a simple procedure, without disturbing the calmness of the patient and causing significant RNA damage. The proposed sampler revealed prolonged SARS-CoV-2 RNA shedding independent of patient severity and from patients with negative nasopharyngeal PCR test results. The assessment of airborne SARS-CoV-2 RNA using the proposed sampler can serve as a basis for the safe discontinuation of COVID-19 isolation precautions, to identify exposure hotspots, and alert individuals at increased risks of infection. To standardize and generalize the sample results obtained from the proposed sampler, long-term validation in different SARS-CoV-2 epidemic situations with different human densities and activities, such as restaurants, schools, and public transportation, such as buses and trains, should be conducted in future research, in addition to comparative studies of airborne and live virus RNA concentrations. Nevertheless, the proposed sampler significantly contributes to an improved understanding of SARS-CoV-2 viral shedding and can facilitate public health interventions, with respect to the prevention and control of the spread of COVID-19.

## Figures and Tables

**Figure 1 microorganisms-11-00944-f001:**
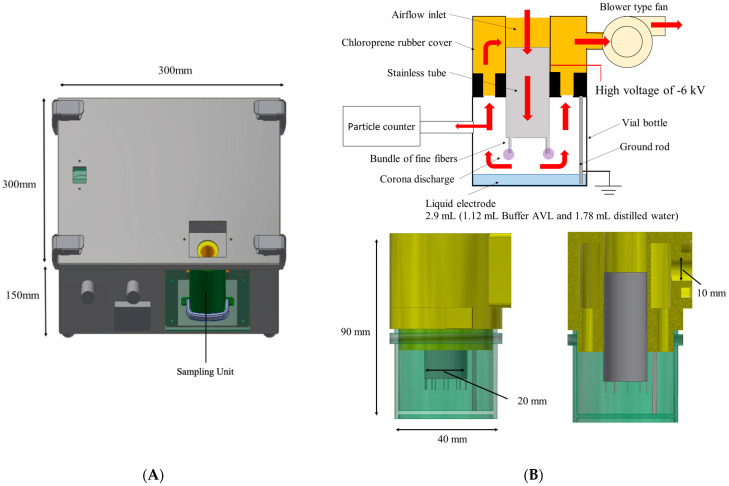
Schematic diagram of the developed electrostatic air sampler. (**A**) Exterior view of the air sampler. (**B**) Schematic diagram of the electrostatic sampling unit. Red arrows indicate the flow of air. The particle counter was installed to measure the particle collection efficiency.

**Figure 2 microorganisms-11-00944-f002:**
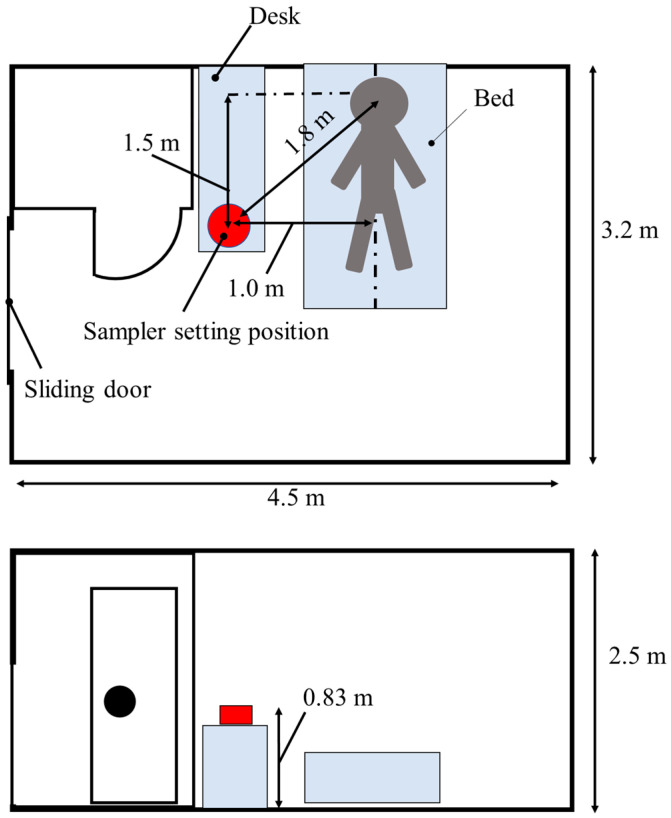
Schematic diagram of the patient room. All patients were admitted to the same single-patient room with dimensions of 4.5 m × 3.2 m × 2.5 m (length × width × height) and an interior volume of 36 m^3^. The sampler was placed 1.8 m from the head of patient and 0.83 m above the floor.

**Figure 3 microorganisms-11-00944-f003:**
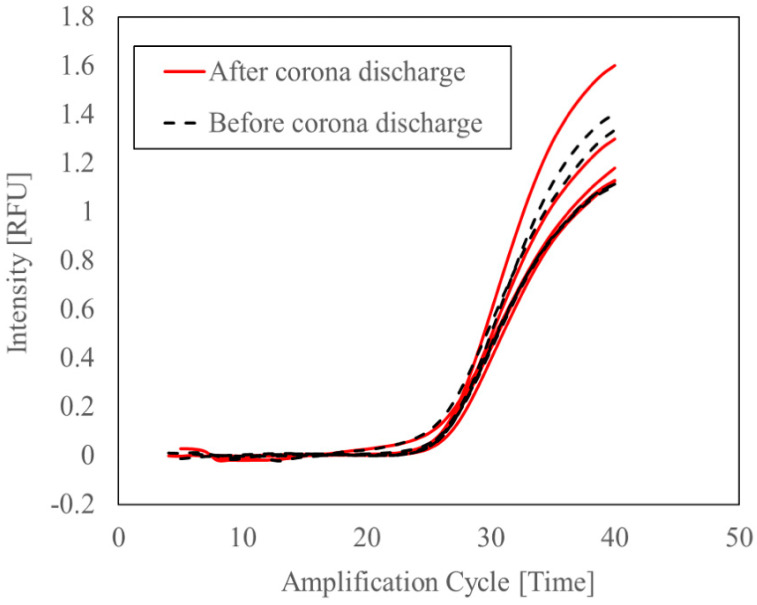
Real-time quantitative PCR (RT-qPCR) threshold (Ct) values of bacteriophage RNA in Buffer AVL before and after exposure to corona discharge, and Ct values of bacteriophage MS2 RNA before (dashed lines) and after (solid lines) exposure to corona discharge were measured five times each.

**Table 1 microorganisms-11-00944-t001:** Clinical characteristics of the patients and SARS -CoV-2 viral RNA concentrations in the hospital room air.

Patient ID	1	2	3	4	5	6	7	8	9	10
Age	80s	80s	30s	50s	70s	60s	80s	30s	30s	30s
Gender	Male	Female	Female	Male	Female	Male	Male	Female	Male	Female
Severity	Severe	Severe	Moderate	Moderate	Moderate	Moderate	Moderate	Mild	Mild	Mild
Variant	Omicron 21K/BA1.1	Delta 21A	Omicron 21K/BA1.1	Omicron 21K/BA1.1	Omicron 21K/BA1.1	Omicron 21K/Ba2	Omicron BA.5	Omicron 21K/BA1.1	Omicron 21K/BA1.1	ND **
Vaccination	Unknown	None	None	2 times	3 times	3 times	Unknown	Unknown	None	None
Ct value *	31.5	27	29.9	21.3	26.3	14.4	17.9	17.6	17.1	ND **
Days after symptom onset	Viral concentration (copies/m^3^)									
0										
1										
2		1.1 × 10^3^				3.0 × 10^2^	3.1 × 10^2^			
3					1.0 × 10^3^			2.4 × 10^4^		
4		(dead)		9.3 × 10^2^	1.0 × 10^3^		0	1.7 × 10^3^		
5				4.7 × 10^3^	1.0 × 10^3^	9.2 × 10^2^	1.8 × 10^2^	0	1.0 × 10^2^	
6			6.0 × 10^2^		0	1.2 × 10^3^		1.2 × 10^2^		
7				1.3 × 10^3^						
8								0	0	
9			0							
10	2.5 × 10^2^		0							3.9 × 10^2^
11	0					(discharge)			(discharge)	
12			(discharge)							(discharge)
13	2.2 × 10^2^			(discharge)						
14										
15										
16								(discharge)		
17	1.4 × 10^2^									
18	1.3 × 10^3^				(discharge)					
19	(dead)									
20										

The gray shaded columns indicate the hospitalization period. * Ct value: cycle threshold values of nasopharyngeal SARS-CoV-2 PCR on admission. ND **: not detected.

**Table 2 microorganisms-11-00944-t002:** RT-qPCR detection results of SARS-CoV-2 in the indoor environment and suspended concentrations.

	Office	Food Court	Station Corridor
Space volume (m^3^)	340	2000	1500
Number of people	30	300	100
m^3^/people	11	6.7	15
Ct value	33.7	36.3	ND **
Virus RNA concentration (copy/m^3^)	7.8 × 10^2^	1.9 × 10^2^	ND **

ND **: not detected.

## Data Availability

All publicly available datasets were analyzed in this study.

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
