# Peer review of "Novel Virus Air Sampler Based on Electrostatic Precipitation and Air Sampling of SARS-CoV-2"

_microorganisms, 2023, doi:10.3390/microorganisms11040944_

Round 1
Reviewer 1 Report
Development of a novel virus air sampler based on electrostatic 2 precipitation and air sampling of SARS-CoV-2.
This is an incredibly important topic, measuring the number of viral particles of airborne particles to determine the safety level of a working environment. I applaud the authors in their attempt to improve on current methods to make them more efficient, and improving on the limitations of electrostatic methods by adding a radical scavenging solution to limit the damage caused to nucleic acids in an electrostatic system.
Still there are some points that have to be discussed further.
I am a little alarmed by the lack of comparison to current methods. Therefore, if the currently used method is vaccum filtration, I believe it is incumbent on the authors to compare that method to the current one. I dont even suggest the authors need to test coronaviruses necessarily in this comparison. Simply releasing defined amounts of bacteriophages into the air, followed by particle counting, vaccuum filtration in tandem with the use of the electrostatic method in a laboratory environment should suffice.
This is important to understand the reliability and reproducibility of the method.
Also, as the authors point out, this method has been employed in the past for influenza viruses, how robust was that method and how do the authors know? How do their results compare.
Also, I believe it is necessary to provide some explanation for why severe patients sometimes gave a lower airborne viral count than mild patients. Does that point to limitations in the method, or do severe patients give off lower airborne viral counts. This should be discussed.
Overall, I given the clear importance of the topic, and the amount of work the authors dedicated, I would tend to recommend publication, but not without a sincere discussion of the above topics and some data indicating the robustness and reproducibility of the method.
Reviewer 2 Report
Dear Authors
After a detailed reading in the manuscript, entitled: “Development of a novel virus air sampler based on electrostatic precipitation and air sampling of SARS-CoV-2”, I realized that this study has a high level of scientific contribution, by developing a new wet-type electrostatic precipitator air sampler using a viral dissolution buffer containing a radical scavenger agent and verified the concentration of severe acute respiratory syndrome coronavirus 2 (SARS) -CoV-2) RNA in the air of a hospital room of coronavirus disease (COVID-19) patients and public areas. Where the results provide observations on SARS-CoV-2 RNA in the air using the sampler developed can be used to guide safe interruption of COVID-19 isolation precautions, to identify exposure hotspots and to alert individuals at increased risk of infection. This is redundant and globally important. Therefore, I suggest ACCEPT the manuscript for publication with minor corrections:
1 - I believe that the word "monitoring" is not appropriate, it could be replaced by terms such as: "assessment", or "lifting".
2 - At the end of the introduction, it would be necessary to report the importance of your study at a global level, extolling the importance of this study.
3 - In the conclusion item, it is necessary to return to the general objective and report the main result. Answering what is the contribution of work to society.
4 - At the end of the conclusion item, you should suggest the possibility of future studies that can continue this study.
5 - References are adequate. Congratulations to the authors.
Reviewer 3 Report
This is a very interesting paper which describes a novel virus air sampler using an electrostatic precipitation with the viral lysis buffer containing the radical scavenger. I believe the method developed by authors is potentially useful for the COVID-19 measures. However, some points should be addressed before acceptance.
1) Authors determined the collection efficiency of the device at the flow rate of 40 L/min and resulted in 46 %. The reliability of this value is very important because it was used to determine the viral RNA concentration in air (Cv). My simple question is how many times this value is obtained repeatedly.
2) As for the collection efficiency, please mention the potential influences of corona discharging conditions (voltage, amount of buffer solution, etc.) and atmospheric condition (temperature and relative humidity).
3) If possible, please describe the content of guanidinium thiocyanate in the buffer solution.
4) The viral concentrations shown in Table 1 and 2 seems valuable for readers. However, these values should be compared with those measured by previous studies for the validation in the discussion section.
Round 2
Reviewer 1 Report
Thank you for this revision. I think this paper is now published.